# Impact of Obesity in Kidney Diseases

**DOI:** 10.3390/nu13124482

**Published:** 2021-12-15

**Authors:** Vasilios Kotsis, Fernando Martinez, Christina Trakatelli, Josep Redon

**Affiliations:** 13rd Department of Internal Medicine, Hypertension-24h ABPM ESH Center of Excellence, Papageorgiou Hospital, Aristotle University of Thessaloniki, 564 29 Pavlos Melas, Greece; vkotsis@auth.gr (V.K.); ctrak@auth.gr (C.T.); 2Internal Medicine Hospital Clínico de Valencia, 46010 Valencia, Spain; fernandoctor@hotmail.com; 3Cardiovascular and Renal Research Group, INCLIVA Research Institute, University of Valencia, 46010 Valencia, Spain; 4CIBERObn Carlos III Institute, 28029 Madrid, Spain

**Keywords:** obesity, fatty kidney, glomerulopathy, CKD, ESRD, bariatric surgery

## Abstract

The clinical consequences of obesity on the kidneys, with or without metabolic abnormalities, involve both renal function and structures. The mechanisms linking obesity and renal damage are well understood, including several effector mechanisms with interconnected pathways. Higher prevalence of urinary albumin excretion, sub-nephrotic syndrome, nephrolithiasis, increased risk of developing CKD, and progression to ESKD have been identified as being associated with obesity and having a relevant clinical impact. Moreover, renal replacement therapy and kidney transplantation are also influenced by obesity. Losing weight is key in limiting the impact that obesity produces on the kidneys by reducing albuminuria/proteinuria, declining rate of eGFR deterioration, delaying the development of CKD and ESKD, and improving the outcome of a renal transplant. Weight reduction may also contribute to appropriate control of cardiometabolic risk factors such as hypertension, metabolic syndrome, diabetes, and dyslipidemia which may be protective not only in renal damage but also cardiovascular disease. Lifestyle changes, some drugs, and bariatric surgery have demonstrated the benefits.

## 1. Introduction

The continuous growing of the obesity pandemic introduced a great bulk of disease beyond the classical well recognized metabolic, orthopedic, psychological, and cardiovascular consequences of obesity. Beside these, although in some ways linked to, is the impact of obesity in the kidney, usually silent during the years before producing relevant damage with clinical expressivity. A review in epidemiology, mechanisms, clinical, and therapeutic aspects is presented.

### 1.1. Epidemiology

In adults within a large, community-based population, in an integrated system of health care delivery in whom serum creatinine had been measured between 1996 and 2000 and who had not been supported with dialysis or undergone kidney transplantation, an independent, ranked association between a reduced estimated GFR and the risk of death, cardiovascular events, and hospitalization was reported. As GFR decreased from 59 to 45 (mL/min/1.73 m^2^), the risk of death increased to 1.8 reaching gradually the highest values of almost 6 times more in the end stage renal disease patients [1]. Similar results for the association of CKD progression with cardiovascular disease and death was also reported in metanalysis studies [2,3]. The unadjusted prevalence of stage 3 and 4 CKD in USA increased from the late 1990s to the early 2000s with a rise in the prevalence of diabetes, hypertension, and obesity. However, the overall prevalence has stabilized since 2003 to 2004 and 6.9% of the population has CKD in 2011 to 2012. Reasons for the recent stabilization of overall CKD prevalence despite continued aging of the U.S. population and the increased prevalence of obesity include better control of hypertension, successful glycemic control with the newer drugs, and expanded use of medications blocking the renin–angiotensin system in patients with proteinuria [4]. In the UK, the prevalence of eGFR < 60 mL/min/1–73 m^2^ was 7.7%, 7.0% and 7.3% in 2003, 2009/2010, and 2016, respectively initially decreased in 2010 but again increased in 2016 [5]. At the same time in the UK the prevalence of diabetes and obesity increased during 2003–2016, while prevalence of hypertension and smoking fell.

Beyond the impact of diabetes and hypertension on the increased risk of CKD in the obese patients, the direct role of obesity in kidney injury has been demonstrated in animal models and epidemiological studies in humans. BMI has been reported as an independently significant factor for the development of CKD in most studies [6,7,8,9,10] with an odds ratio from the lowest to the highest BMI of 1.273 for developing ESKD, after adjustment for age, sex, systolic blood pressure, and proteinuria [6], but obesity risk appeared largely mediated by diabetes and hypertension in other studies [8] or not to be associated with ESKD [11]. A recent metanalysis of studies from a general population with normal baseline renal function reported that obesity increased the relative risk of developing low eGFR by 1.28 and albuminuria by 1.51 [12]. Table 1 collected studies of CKD risk in adults with metabolic syndrome.

Nondiabetic participants with normal baseline kidney function and metabolic syndrome according to the National Cholesterol Education Program who were included in the Atherosclerosis Risk in Communities study had an adjusted risk of developing CKD of 1.43 compared with participants with no features of the metabolic syndrome [19]. Central fat distribution indices, i.e., waist to hip circumference or visceral fat, are better related with the risk of ESKD compared to BMI [26,27,28]. Meta analyses reported that metabolic syndrome components such as obesity, impaired fasting glucose, elevated blood pressure, and hypertriglyceridemia were associated with significant increases in proteinuria and albuminuria risk [25].

Metabolically healthy obesity (MHO) is an obesity phenotype that obesity is not associated with metabolic complications such as insulin resistance, inflammation, hypertension, or T2D. Compared with metabolically healthy non-obesity phenotype, the odds ratios for incident CKD for MHO were similar to the comparison group, but significantly increased for metabolically abnormal non-obese and obesity phenotype respectively after adjustment for confounders [29,30,31]. However, other studies reported that MHO may have an intermediate future risk to develop ESKD.

### 1.2. Pathology

Obesity-Related Glomerulopathy

Physical compression of the kidneys from accumulation of adipose tissue around the organs emphasizes the possible role of visceral obesity in the development of renal disease. Deposition of extracellular matrix throughout the renal medulla is expanded and the tissue surrounding the ducts of Bellini at the vascular pole tends to prolapse. Increased numbers of interstitial cells and material rich in lipids and proteoglycans press the renal parenchyma towards the pole of the kidney resulting in the formation of round-shaped, enlarged kidney in obese subjects. Renal compression affects both vascular (the vasa recta) and tubular (the Henle’s loops) elements causing activation of the RAS and increased sodium reabsorption [32,33,34].

The primary histologic features are few lesions of focal-segmental glomerulosclerosis, profound glomerulomegaly due to glomerular hyalinosis and fibrosis, as well as lipid accumulation in the glomeruli and adhesion to Bowman’s capsule [35,36]. Altered fat metabolism in the kidneys induces lipid accumulation, suggesting that high fat intake may have a direct lipotoxicity effect in the kidneys. Ectopic lipid accumulation in the kidneys induces structural and functional changes of the mesangial cells, podocytes, and proximal tubular cells. Perivascular fat in the renal sinus appears to participate in vascular function, modifying the blood flow in the underlying arteries. Obesity increases renal mass and glomerular diameter. Podocytes need to enlarge their processes to cover an expanded area that cause podocyte detachment, loss in protein selectivity, formation of denuded areas that trigger matrix deposition, and podocyte damage.

Glomerular changes in obesity-induced renal injury are unmatched to those of diabetic nephropathy, due to the lower severity in the first in the mesangial space changes. Patients with diabetic nephropathy more frequently have albuminuria, proteinuria, and ESKD compared to obesity induced nephropathy where the results are slower and the progression to end renal disease is less frequent. Other causes of renal injury, apart from high fat intake, could include overexpression of Ang II with a consequent increase in proliferative factors such as transforming growth factor (TGF-β) and plasminogen activator inhibitor and insulinemia giving genesis to cell growth. Hyperfiltration because of sodium reabsorption, increase the blood flow to the kidney causing gradual glomerular wall sclerosis due to physical shear stress and a dangerous circle starts in which nephrons are injured leading to their apoptosis, sodium retention attenuates, while blood pressure increases to maintain sodium balance. Proteinuria of nephrotic range is rare among the obese, but albuminuria may exist [35,37]. These findings suggest that obesity-related renal damage should be defined as a special form of focal-segmental glomerulosclerosis slowly progressing to end stage renal disease. Patients with the metabolic syndrome have high prevalence of microvascular disease manifested as tubular atrophy, interstitial fibrosis, and arterial sclerosis [38].

### 1.3. Mechanisms

The mechanisms of structural abnormalities of CKD are related to the obesity-comorbidities, i.e., hypertension, insulin resistance, type 2 diabetes, and atherogenic dyslipidemia, that contribute to renal damage through mechanisms that include inflammation, oxidative stress, RAAS upregulation, increased SNS activity, and endothelial dysfunction that finally induce renal damage [39], Figure 1.

A. Hemodynamics

Activation of the sympathetic nervous system (SNS) has been thought to play an important role in the pathogenesis of hypertension and CKD among obese individuals [40]. Plasma renin activity displayed significant increase in obesity and local perivascular adipose tissue angiotensin II is also increased [33]. Angiotensin II raises the efferent arteriole tone in the glomerulus, production of TGF-beta, fibrosis, and apoptosis of the podocytes. In the early stages of kidney damage associated to obesity, e-GFR is increased due to the hyperperfusion from volume overload. Intrarenal increased physical forces, generating from fat accumulation around and into the renal medulla, diminish flow rate of the filtrate at the loop of Henle and sodium retention is observed [33]. These early changes can be reverted by weight loss, salt restriction, and renin-angiotensin system blockade.

B. Inflammation

Chronic low-grade inflammation develops locally in the expanding adipose cells from macrophage but becomes systemic through the release of pro-inflammatory mediators that include cytokines into the blood stream. Elevated levels of free fatty acids (FFAs) in obese individuals may enhance vascular a-adrenergic sensitivity, inhibit Na^+^, K^+^-ATPase and the sodium pump increasing vascular smooth muscle tone and vascular resistance, activate epidermal growth factor receptor, and produce reactive oxygen species and protein kinase C. A variety of biologically active cytokines are produced in adipose cells, including reactive oxygen species, proinflammatory and inflammatory molecules (interleukin-1β, interleukin-6, tumor necrosis factor-α, C-reactive protein), angiogenetic factors (vascular endothelial growth factor), hemostasis modulating compounds (plasminogen activator inhibitor-1, thromboxane A2), acute phase reaction proteins (serum amyloid A proteins, C-reactive protein), and activation of nuclear factor kappa-light-chain-enhancer of activated B cells (NF-κB) and IκB kinase (IKK), pathways that promote endothelial dysfunction and microvascular disease [33]. Different pathophysiological mechanisms may contribute to the development of CKD from gut microbiota dysbiosis such as production of uremic toxins mainly trimethylamine-N-oxide (TMAO), reduced prophylactic short-chain fatty acids, enhanced inflammation and immune response, reduced nitric oxide (NO), and peptides that block the angiotensin-I converting enzyme [41].

C. Hormones

Insulin resistance induces glomerular hyperfiltration, endothelial dysfunction, increased vascular permeability, angiogenesis, and other pathways implicated in microvascular damage that is associated with albuminuria [42]. Hyperglycemia activates pathways that increase production of advanced glycation end-products (AGEs), activate protein kinase C isoforms, and increase transforming growth factor β that enhances extracellular matrix production by mesangial cells inducing renal fibrosis [43]. Podocytes block proteinuria through arrangement of actin cytoskeleton in their foot processes. Decreased podocyte number and podocyte foot process effacement have been reported in diabetic patients with early phases of kidney damage. Insulin action in podocytes is critical for the glomerular function and structure affecting morphology, cytoskeleton remodeling, and finally their survival [44].

Leptin is a small peptide hormone that is produced in adipose tissue and increases in the blood of obese subjects. The circulating leptin associates with adipose tissue mass and regulates food intake through its hypothalamic actions that release other neurotransmitters. Leptin can modify insulin actions, induce angiogenesis, reduce endothelial NO synthase, and interact with the immune system. Leptin is cleared by the kidney and is increased in patients with chronic renal failure associated with anorexia and weight loss in ESKD patients. Leptin triggers glomerular endothelial cells secretion of TGF-beta, to which sensitized mesangial cells may respond inducing development of focal glomerulosclerosis and proteinuria [45]. Additional effects of leptin on the kidney include natriuresis, increased sympathetic nervous activity, and stimulation of reactive oxygen species [33].

Adiponectin is an adipose tissue derived peptide hormone, reduced in obese subjects, that acts as lipolytic factor and regulates insulin sensitivity. Adiponectin prevents the atherogenic process by inhibiting foam-cell formation. Adiposity is characterized by adiponectin deficiency. Plasma adiponectin levels are inversely related to insulin levels. Adiponectin knockout mice demonstrate a diet dependent insulin resistance and atherogenesis [33]. Adiponectin increases AMPK activity, reducing podocyte permeability [46,47]. Finally, resistin, an inflammatory adipokine produced by the monocyte macrophage cells, is increased in patients with low GFR [48]. In adults with hypertension and diabetes, circulating resistin levels were associated with reduced estimated glomerular filtration rate and albuminuria [49].

### 1.4. Endothelial Dysfunction and Changes in Vascular Structure

Endothelial dysfunction plays an important role in the pathogenesis of CKD and albuminuria [50]. Insulin resistance, low levels of adiponectin, high plasma leptin, increased levels of plasma glucose, and FFAs induce an inflammation profile that causes endothelial dysfunction, which causes increased protein loss from the kidneys.

Nitric oxide (NO) is produced from the endothelium, promotes vasodilation, reduces inflammation, and platelet aggregation. Phosphoinositide 3-kinase activation is causing phosphorylation of endothelial NO synthase (e-NOs) that produce NO [51]. Obesity is associated with reduced NO bioavailability. In the presence of insulin resistance this pathway is down-regulated, while hyperinsulinemia increases endothelin-1 levels resulting in imbalance between vasodilator and vasoconstrictor endothelium factors causing hypertension [52,53]. Vascular cell adhesion molecule-1 (VCAM-1), inter-cellular adhesion molecule-1 (ICAM-1), and E-selectin increase monocyte adhesion to the vascular wall, inducing atherosclerosis. These vascular factors promote a cycle that is causing renal damage and hypertension [33].

## 2. Clinical Consequences

The clinical consequences of obesity on the kidneys, with or without metabolic abnormalities, involve both renal function and structures, Figure 2. Higher prevalence of urinary albumin excretion, sub-nephrotic syndrome, nephrolithiasis, increased risk of developing CKD, and progression to ESKD have been identified as being associated with obesity and having a relevant clinical impact. In renal replacement therapy and kidney transplantation, both the availability of donors and graft survival are also influenced by obesity. At the time of estimating the association and the impact of obesity on renal disease, the presence of sarcopenia, a not infrequent condition, can be misleading, since it can lead to underestimating obesity [54]. Therefore, other parameters beyond BMI should be considered [55].

### 2.1. Urinary Albumin Excretion and Proteinuria

In obese subjects, albuminuria is more frequent. A significant association of albuminuria with either obesity or central obesity has been reported [56,57,58], being higher in the presence of central obesity. The presence of a cluster of cardiovascular risk factors increases the risk [59].

Albuminuria associated with obesity has been observed in children and adolescents. In moderate obese adolescents, the prevalence was reported at 2.4% [60]; however, in severe obesity, 3% displayed proteinuria, 14% microalbuminuria, and 3% had a GFR <60 mL/min/1.73 m^2^ [61]. In addition, Goknar et al. [62] reported that severely obese children had a higher number of urinary markers for tubular damage, such as N-acetyl-beta-D-glucosaminidase (NAG), and kidney injury molecule (KIM)-1.

Even though the prevalence of albuminuria in the presence of obesity has been demonstrated, this condition remains underdiagnosed due to the absence of clinical symptoms and lack of specific search of low-grade albuminuria.

### 2.2. Sub-Nephrotic Syndrome

Obesity-related glomerulopathy is a characteristic syndrome which is categorized by the presence of sub-nephrotic proteinuria, glomerulopathy, and renal function loss. Patients usually do not have proteinuria at the level of nephrotic syndrome in 30% of subjects [63] in which there is sub-nephrotic proteinuria in the absence of edema, hypoalbuminemia, and less hyperlipidemia. The reason for the differences between this syndrome and typical nephrotic syndrome is the indolent development of compensating mechanisms over many years. These mechanisms reduce or limit the systematic and metabolic impact, increasing hepatic synthesis of albumin and other proteins [64]. This is in contrast with nephrotic syndrome due to other etiologies. Biopsies in obese patients reveal glomerulomegaly and some of them also develop an adaptive form of focal segmental glomerulosclerosis, increasing the risk of progression to renal dysfunction [65].

### 2.3. Progression to CKD and ESRD

Obesity has been associated with a higher incidence of CKD defined by the presence of albuminuria and/or GFR < 60 mL/min/1.73 m^2^ as compared to the non-obese population [12,66,67,68]. The impact of obesity on conditions that favor the progressive decline of renal function has been emphasized. Low birth weight children, low renal endowment, subjects with reduced renal mass due to different origins, or with primary or secondary renal damage, displayed an increased risk of progression toward CKD and ESKD in the presence of obesity [69]. The role of metabolic abnormalities obesity-associated in the increment of risk has received attention. While some studies support that the metabolically healthy obese (MHO) do not have an increased risk of progression toward CKD and ESKD [70,71], or even a reduction in risk [72]. However, other studies are more in favour of MHO being the first stage of obesity [73] and that it is a question of time as to the development of metabolic abnormalities and consequently an increased risk of the development of renal dysfunction.

Individuals who are obese have a more than 3-fold higher risk of developing end-stage kidney disease (ESKD) than those with normal bodyweight [74,75]. In a large cohort from Austria, with a prevalence of obesity of 11.8%, 0.3% developed ESKD in a follow-up of 22 years and an increase of 5 points of BMI increased the risk by 56% [76]. In a cohort of the Kaiser Permanent register with 320,252 subjects followed over 21 years, the hazard risk for ESKD increased through the obesity grade 3.57, 6.10, and 7.07 for obesity 1 to III respectively, as compared with normal weight subjects [74]. However, when the rate of decline of renal function in CKD to develop ESKD was evaluated, controversial data had been reported. While some studies reported a faster decline [77] in the presence of obesity, other did not confirm [78].

### 2.4. Nephrolithiasis

Prevalence and incidence of nephrolithiasis is increased in obese subjects. Association is facilitated by lower urinary pH, increased urinary oxalate, sodium and phosphate excretion, and uric acid. Other factors such as the effect of insulin resistance on tubular H-Na exchanger and ammoniagenesis promoting urine acidification have also been implicated in the pathogenesis [79]. It is worth commenting that the risk increases after certain weight loss therapies. In fact, after Roux-en-Y, gastric bypass absorption of oxalate in the intestine largely increases and the risk of nephrolithiasis needs to be prevented by reducing dietary oxalate consumption and oral calcium supplementation.

### 2.5. Renal Replacement Therapy

The increasing prevalence of obesity produces a challenge for optimal care of patients in renal replacement therapy, in both hemo- and peritoneal dialysis [80]. In the case of hemodialysis, at 3 years, adiposity of the subcutaneous tissue produced problems with vascular access and a reduction in the catheter functionality. Moreover, in obese subjects, increased dialysis time or frequency is necessary, and it is more difficult to achieve the dry weight. Proximal calciphylaxis is more frequent in obese than in lean patients. In patients on peritoneal dialysis, catheter malfunction and exit site infections are more prevalent in obese subjects. In some patients with severe adiposity, a prophylactic omentectomy could be useful. In addition, patients with advanced CKD, especially those undergoing dialysis, tend to have severe nutritional disorders, protein-energy wasting, and the presence of obesity may be better in this population, an obesity paradox [81].

### 2.6. Kidney Transplantation

In the past, obesity was a contraindication for kidney transplantation if weight was not reduced. Despite the fact that cut-off limits have increased, even until a BMI of 40 kg/m^2^, obesity is still one of the leading causes of being inactive on the transplant list. The reason for this is that obese subject recipients of transplantation have increased rates of delayed graft function, wound infection, and rejection.

It is also relevant to note the impact of obesity on the living kidney donor pool and the acceptance of organs from obese subjects. In the former, there is a risk for the donor and recipient, since a mass reduction in an obese subject puts them at risk for future ESKD and in the latter, delayed graft function is more frequent if the donor is obese. According to KDIGO recommendations, an individualized decision should be made for a living donor if the BMI > 30 kg/m^2^ due to the risk for future development of hypertension, diabetes as well as ESKD [82].

### 2.7. Renal Cancer

Obesity has been associated with an increased risk of kidney malignancy. Several studies have concluded the increment of risk associated with obesity and it has been estimated that 20% of renal cancer patients were obese. The risk of kidney cancer was increased by 35% in overweight participants and by 76% in obese subjects in comparison to normal weight participants, irrespective of the gender [83]. The association is consistent in both sexes and across populations; however, no clear explanation for the pathogenesis has been found.

### 2.8. Fatty Kidney

Accumulation of ectopic fat in the kidney is receiving more attention in the last years and will increase with the development of techniques that allow for a better estimation than the classical echography and CT-scan. Apart from intra-renal accumulation in the proximal tubule and in minor grade in the glomeruli, fat in the renal sinus and around the renal capsule seems to play a role in the renal dysfunction of the obese patient. Renal sinus fat has been associated with CKD in the Framingham Heart Study [84]. Moreover, perirenal fat seems to produce lipotoxic effects on the kidney, increasing the glomerular hydrostatic pressure and the activity of the renin-angiotensin-aldosterone system, contributing to the progression of kidney damage [85].

### 2.9. Other Obesity-Associated Conditions and Renal Damage

Two frequent complications of obesity seem to further increase the risk of renal damage. First is sleep-apnea and nocturnal hypoxemia, which have been associated with loss of kidney function through activation of the renin-angiotensin system [86]. Second is non-alcoholic fatty liver disease (NAFLD). In a meta-analysis of 33 studies, NAFLD, non-alcoholic steatohepatitis, and advanced fibrosis were associated with an increased risk of prevalence and incidence of CKD, with a graded risk from the presence to the severity of NAFLD [87].

## 3. Treatment of Obesity and Renal Damage

Losing weight is key in limiting the impact that obesity produces on the kidneys by reducing albuminuria/proteinuria, declining rate of eGFR deterioration, delaying the development of CKD and ESKD and improving the outcome of a renal transplant. The resulting effects due to weight reduction are multiple. Aside from a reduction in BP as well as control of other CV risk factors, reduction of leptin, glomerular hyperfiltration, RAAS activity, inflammation, and oxidative stress seem to be the most relevant. Considering the characteristic hyperfiltration hemodynamic profile and the relevance of hyperfiltration-mediated conditions in obesity-induced renal damage, reduction in filtration fraction is the main mechanism that provides a beneficial impact to the subject who has lost weight. In addition, a reduction in the activity of the RAAS has also been observed [88]. Weight reduction may also contribute to appropriate control of cardiometabolic risk factors such as hypertension, metabolic syndrome, diabetes, and dyslipidemia which may be protective not only in renal damage but also cardiovascular disease [89].

### 3.1. Life Style Intervention

Patients with obesity, particularly those with markers of renal injury (albuminuria/tubular markers or eGFR < 60 mL/min/1.73 m^2^), need to be encouraged to lose weight through a combination of diet and physical exercise. If addressed early, a low-calorie diet, with or without physical exercise, is able to reduce albuminuria with a decrease being proportional to the reduced weight. Weight loss achieved through a combination of diet and exercise has also had beneficial effects on the reduction in urinary protein excretion. After diet introduction, it is possible to observe UAE reduction in a few weeks. In a control trial, which lasted for five months, a 4% weight reduction decreased proteinuria in around 50% of subjects [90]. However, data on slowing the progression to CKD were less documented due to difficulties in assessing the outcomes [91,92] and the short-term duration of the studies [55].

A low-calorie diet with salt restriction is recommended since it can contribute to BP reduction [55]. Further salt intake reduction should be implemented if proteinuria is present. Addition of fibre in the diet promotes growth of short-chain fatty acid producing bacteria that have been demonstrated to reduce all-cause mortality in CKD [93] and there seem to be promising results regarding preclinical CKD risk [94]. A high-protein diet is not recommended since it increases GFR and UAE.

A recent manuscript reviewed the randomized clinical trials of lifestyle intervention in patients with CKD [55]. Diet intervention with low-calorie and salt restriction reduce weight and albuminuria; however, no convincing data exist for other specific dietary pattern such as low fat, low carbohydrate, or a Mediterranean diet. Studies performed on the impact of physical exercise demonstrated a reduction in BP, BMI, and improvement in exercise capacity and quality of life; however, no reduction in albuminuria was observed. The limitation of losing weight with lifestyle is that the maximal reduction achieved is 3 to 4% and the maintenance overtime is poor, therefore other additional actions need to be implemented.

### 3.2. Medications

RAAS blockers

In the presence of albuminuria or proteinuria, RAAS blockers should be prescribed to reduce not only the overactivity of the system but also the sympathetic overactivity, HTN, insulin resistance, and low-grade inflammation. The most important effect is to reduce the filtration fraction and consequently albuminuria; however, in CKD patients the reduction in eGFR should be monitored after starting treatment.

Antiobese-drugs

Of the drugs approved for treatment of obesity, Phentermine-Topiramate, GLP-1 receptor agonist, and Bupropion-Naltresone, mainly data about the impact on renal function is available for the GLP1 agonist. This class of drug has been tested for renal protection in diabetic patients. Liraglutide, a GLP1 agonist, introduced initially as a glucose-lowering drug with impact in body-weight, is able to reduce weight and in a recent trial, LEADER, demonstrated reduction in CV risk [95]. A significant decrease in albuminuria, new onset of persistent proteinuria, and no progression of eGFR decline have been reported in diabetic patients. SUSTAIN-6 with semaglutide [96], another member of the GLP1 agonist class, reduces the risk of composite renal outcome largely driven by persistent proteinuria. However, Dulaglutide in the AWARD-7 [97] did not find differences in the reduction of albuminuria. Topiramate in one study did not demonstrate a beneficial impact on renal outcomes in Type 2 diabetes. Lorcaserin, a selective serotonin 2C receptor, in high cardiovascular risk patients, offered a reduced rate of renal impairment in comparison with placebo [98]. The beneficial impact of GLP1 is that it may protect the kidney from progression to CKD and/or ESKD.

Sodium-glucose cotransporter 2 inhibitors

Sodium-glucose cotransporter 2 inhibitor (SGLT2i) is a class of drug released in the last years with a mechanism that produces various beneficial effects in patients with diabetes, obesity, and cardio and renal protection. Inhibition of glucose reabsorption in the proximal tubule produce glucosuria, reducing the caloric burden and the sodium content with reduction of blood volume, and increasing the sodium arrival to the yuxtaglomerular corpus, inhibiting the hyperactivity of the renal angiotensin system and reducing the filtration fraction, giving protection to the kidney [99]. As a consequence, slightly reduced weight, reduced BP, and the GFR that results can protect of renal function at large. Additional mechanisms in NH3, sympathetic activity, and oxidative stress with beneficial effects complete a frame of a very useful drug. Several outcome trials supported the beneficial impact of the drug in the cardiovascular and renal outcomes [100,101,102,103]. The last European of Society of Cardiology and the European Society of Diabetes (ESC/EASD) recommended introduction of SGLT2i in the first step of diabetic patients with very high risk or previous cardiovascular events [104]. In patients with obesity associated diabetes, it may be a good choice in order to protect renal function. The same applies in obese subjects with increased urinary albumin excretion or proteinuria. However, in patients with reduced GFR the efficacy of the drug is reduced and the impact on protection is diminished. As GFR < 45 mL/min/1.73 m^2^ is almost negligible, it is a challenge to their use. The beneficial impact observed in patients with GFR between 30–45 mL/min/1.73 m^2^ in trials and the absence of side effect point to their use off-label [105].

### 3.3. Bariatric Surgery

Bariatric surgery, also so-called metabolic surgery, refers to methods used to reduce obesity and improve metabolic abnormalities. The most used techniques are the vertical sleeve gastrectomy, Roux-en-Y gastric bypass, adjustable gastric banding, and biliopancreatic diversion/duodenal switch. Selection of the most appropriate procedure needs to consider not only the morbidity/mortality risk during the procedures but also the potential side effects during follow-up [82]. The surgical treatment of obesity improved management of diabetic and non-diabetic CKD and reduced the rate of renal decline toward ESKD. Once ESKD is established, absolute event rates are low and although complications can be present, it remains a safe intervention. Implementing one of the above surgeries pre-transplant increases the potential access to transplantation and, in addition, improves the management of metabolic complications post-transplantation, including new-onset diabetes. Likewise, this may be beneficial as a treatment for potential obese donors [82].

A beneficial impact on obese patients with type 2 diabetes in kidney protection has been emphasized. A large metanalysis concluded that post-surgical reduction in albuminuria is independent of the changes in BMI, HbA1c, and systolic BP [106]. In non-diabetic subjects, randomized studies are not available, but many observational studies have demonstrated the beneficial impact in reducing incidence of albuminuria and the risk for ESKD after an 18 year follow-up [107]. In CKD patients, at the end of the first year post-surgery [108] and after 7 years of follow-up, improvement in the categories of CKD were observed in around half of the patients, and even in patients with very high risk at baseline a quarter of them improved [109].

In patients with ESKD in dialysis, before kidney transplantation, bariatric surgery had a reduction in mortality, incidence of diabetes, and around 60% of cardiac diseases. The preferred method in dialysis patients is the laparoscopic sleeve gastrectomy in which results are more effective with less complications and with the additional advantage of not interfering with pharmacokinetics of immunosuppression drugs [110].

Increment in the incidence of kidney stones has been reported associated with bariatric surgery. Among the factors that contribute to this association are the decrease in urinary volume and citrate, the increased urinary oxalate, and the calcium oxalate saturation. Procedure selection may be critical to mitigate the risks of oxalate nephropathy since more restrictive procedures reduce the risk [111].

## 4. Conclusions

The impact of obesity on the kidney has received attention after the recognition that BMI is the second most important marker in developing ESKD after proteinuria and one of the most relevant associated with the presence of CKD, since obesity is frequently associated with hypertension, metabolic syndrome, and diabetes. An important impact on subjects with renal replacement therapy and renal transplantation is also present. The pathologic lesions included a characteristic glomerulopathy coupled with cellular fat load and perivascular fat deposit as well as so-called fatty kidney with fat deposits in the perirenal and renal sinus. The mechanisms linking obesity and renal damage are well understood, including several effector mechanisms with interconnected pathways. In the presence of an increment in urinary albumin excretion, it is mandatory to take action in order to reduce overweight and to control hypertension, diabetes, and dyslipidemia to further prevent GFR reduction.

## Figures and Tables

**Figure 1 nutrients-13-04482-f001:**
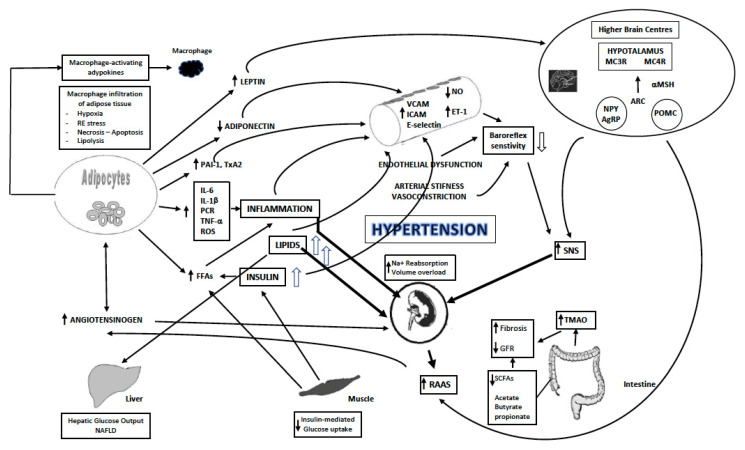
Mechanisms of obesity induced renal damage. Modified from Kotsis V et al., J Hypertens. 2018 Jul; 36(7):1427–1440.

**Figure 2 nutrients-13-04482-f002:**
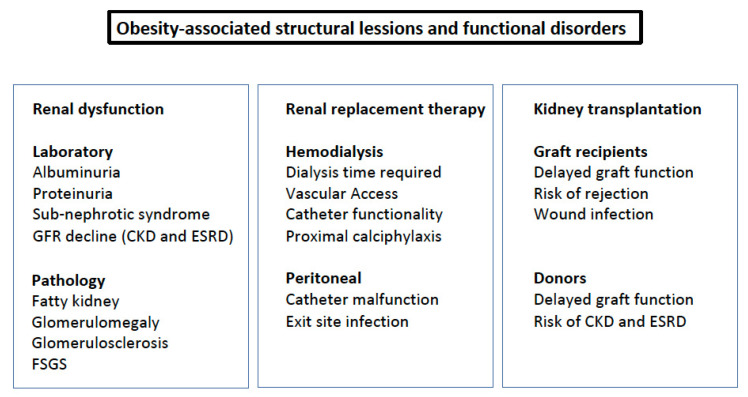
Obesity-associated structural lesions and functional disorders. FSGS: focal segmentary glomerulosclerosis.

**Table 1 nutrients-13-04482-t001:** Risk of CKD in adults with metabolic syndrome (MS).

Study	Patients	Study Endpoint	OR (CI 95%) for Chronic Kidney Disease
Chen et al. [13]	6217 US adults	chronic kidney disease and microalbuminuria	2.60 (1.68–4.03) versus non metabolic syndrome
Palaniappan et al. [14]	6217 American adults	microalbuminuria	OR 2.2 (1.44–3.34) and 4.1 (2.45–6.74) for women and men versus non metabolic syndrome.
Chen et al. [15]	15,160 Chinese adults	chronic kidney disease	1.64 (1.16–2.32) versus non metabolic syndrome
Tanaka et al. [16]	6980 Japanese adults	chronic kidney disease	<60 years; 1.69 (i.35–2.11) versus non metabolic syndrome
Chang et al. [17]	60,921 Korean adults	chronic kidney disease	1.68 (0.57–1.80) versus non metabolic syndrome
Ryu et al. [18]	10,685 Korean healthy men/40,616.8 person-years	prospective, chronic kidney disease	2.00 (1.46–2.73)
Kurella et al. [19]	10,096 US adults/9 years of follow-up	prospective, CKD	1.43 (1.18–1.73)
Yang et al. [20]	4248 Chinese adults/5.40 years of follow-up	prospective chronic kidney disease	1.42 (1.03–1.73)
Ninomiya et al. [21]	1440 adults/5 years of follow up	prospective, chronic kidney disease	2.08 (1.23–3.52)
Lucove et al. [22]	1484 Native Americans/10 years follow up	prospective, chronic kidney disease	1.3 (1.10–1.60)
Sun et al. [23]	118,924 Taiwanese/3.7 years follow up	prospective, chronic kidney disease	1.30 (1.24–1.36)
Chen J [24]	26,601 subjects	chronic kidney disease	1.99 (1.57–2.53)
Rashidbeygi E et al. [25]	10,603,067 participants	meta-analysis albuminuria and proteinuria	1.92 (1.71–2.15) and 2.08 (1.85–2.34)
Thomas et al. [24]	30,146 adults	meta-analysis chronic kidney disease	1.55 (1.34–1.80)

## Data Availability

Not applicable.

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
