# Peer review of "Impact of Obesity in Kidney Diseases"

_nutrients, 2021, doi:10.3390/nu13124482_

Round 1
Reviewer 1 Report
This is an interesting and comprehensive review about obesity-related kidney diseases. However, there are several concerns.
(1) This review consists of ‘1. Introduction’, ‘2. Clinical consequences’, and ‘3. Treatment of obesity and renal damage’. I think that the title of 1. Introduction is not appropriate. Considering the content, the authors should change the title. (2) End-stage kidney disease (ESKD) is preferable to ESRD (Kidney International 97: 1117–1129, 2020). Full spelling of all abbreviations is necessary. (3) (Table 1): More detailed information is required. At least, odds ratio (OR) (95% confidential interval) of each report should be necessary. How about the OR of urinary albumin excretion (UAE)? There were two endpoints (chronic kidney disease: CKD and UAE) in this report (Ref. 18). In prospective observational cohort study, the follow-up period of each study is also required. (4) ‘2. Clinical consequences’ 2.3 ‘Progression to CKD and ESRD’: Some parts are overlapping with 1. Introduction’ ‘Epidemiology’ and ‘Table 1’. I recommend to focus on ‘progression to ESRD (ESKD)’ in this section. (5) ‘2. Clinical consequences’ 2.5 ‘Renal replacement therapy’: As the authors described, obesity-induced troubles of vascular access and catheter function are important. However, the patients with advanced CKD, especially those undergoing dialysis, tend to have severe nutritional disorders (protein-energy wasting: PEW). Considering the prognosis, obesity may be better in this population (as known as ‘obesity paradox’). The authors should adequately discuss these points. (5) ‘3. Treatment of obesity and renal damage’ ‘Medications’: The authors referred to GLP-1 receptor agonists. Rather, sodium-glucose cotransporter (SGLT2) inhibitors are more relevant to obesity-related kidney diseases. They have to discuss SGLT2 inhibitors considering obesity. (6) Figures are smudgy. Correction of figures is necessary.
Author Response
REVIEWER 1
Thanks to the reviewer for his detailed and useful review
This is an interesting and comprehensive review about obesity-related kidney diseases. However, there are several concerns. This review consists of ‘1. Introduction’, ‘2. Clinical consequences’, and ‘3. Treatment of obesity and renal damage’.
(1)I think that the title of 1. Introduction is not appropriate. Considering the content, the authors should change the title.
Title has been modified according with the reviewer:
“IMPACT OF OBESITY IN KIDNEY DAMAGE”
(2) End-stage kidney disease (ESKD) is preferable to ESKD (Kidney International 97: 1117–1129, 2020).
ESKD replace ESRD
Full spelling of all abbreviations is necessary.
This has been done
(3) (Table 1): More detailed information is required. At least, odds ratio (OR) (95% confidential interval) of each report should be necessary. How about the OR of urinary albumin excretion (UAE)? There were two endpoints (chronic kidney disease: CKD and UAE) in this report (Ref. 18).
OR is already reported. 95% CI of odds ratio are available in the original research articles.
(4) In prospective observational cohort study, the follow-up period of each study is also required.
Time of follow-up is already reported. 95% CI of odds ratio are available in the original research articles.
(5) Clinical consequences. 2.3 ‘Progression to CKD and ESKD’: Some parts are overlapping with 1. Introduction’ ‘Epidemiology’ and ‘Table 1’. I recommend to focus on ‘progression to ESKD (ESKD)’ in this section.
ESKD was used across the manuscript for your advice
The advice about center in this section in the risk to develop ESKD has been take in to account and the paragraph about risk has been modified.
Page 10. “Individuals who are obese have a more than 3-fold higher risk of developing end-stage kidney disease (ESKD) than those with normal bodyweight (74,75). In a large cohort from Austria, with a prevalence of obesity of 11.8%, 0.3% develop ESKD in a follow-up of 22 years and an increase of 5 points of BMI increased the risk 56% (76). In a cohort of the Kaiser Permanent register with 320,252 subjects followed 21 years, the hazard risk for ESKD increases through the obesity grade 3.57, 6.10 and 7.07 for obesity 1 to III respectively, as compared with normal weight subjects (74). However, when the rate of decline of renal function in CKD to develop ESKD was evaluate controversial data had been reported. While some studies reported a faster decline (77) in the presence of obesity, other did not confirm (78).
- Hsu CY,McCulloch CE, Iribarren C, Darbinian J, Go AS. Body mass index and risk for end-stage renal disease. Ann Intern Med. 2006;144:21-28. doi:10.7326/0003-4819-144-1-200601030-00006
75.Zitt E, Pscheidt C, Concin H, Kramar R, Lhotta K, Nagel G. Anthropometric and metabolic risk factors for ESRD are disease-specific: results from a large population-based cohort study in Austria. PLoS One. 2016;11:
e0161376. doi:10.1371/journal.pone.0161376
- Fritz J, Brozek W, Concin H, Nagel G, Kerschbaum J, Lhotta K, Ulmer H, Zitt E. The Triglyceride-Glucose Index and Obesity-Related Risk of End-Stage Kidney Disease in Austrian Adults. JAMA Netw Open. 2021;4:e212612. doi:
10.1001/jamanetworkopen.2021.2612.
- Lu JL, Molnar MZ, Naseer A, Mikkelsen MK, Kalantar-Zadeh K, Kovesdy CP. Association of age and BMI with kidney function and mortality: a cohort study. Lancet Diabetes Endocrinol. 2015;3(9):704-714. doi:10.1016/S2213-
8587(15)00128-X
- Swartling O, Rydell H, Stendahl M, Segelmark M, Trolle Lagerros Y, Evans M. CKD Progression and Mortality Among Men and Women: A Nationwide Study in Sweden. Am J Kidney Dis. 2021 Aug;78(2):190-199.e1.
- Clinical consequences’ 2.5 ‘Renal replacement therapy’: As the authors described, obesity-induced troubles of vascular access and catheter function are important. However, the patients with advanced CKD, especially those undergoing dialysis, tend to have severe nutritional disorders (protein-energy wasting: PEW). Considering the prognosis, obesity may be better in this population (as known as ‘obesity paradox’). The authors should adequately discuss these points. (5) ‘
A reference to the commentary has been included
Page 10. “In addition, patients with advanced CKD, especially those undergoing dialysis, tend to have severe nutritional disorders, protein-energy wasting, and the presence of obesity may be better in this population, an obesity paradox
- Kittiskulnam P, Johansen KL. The obesity paradox: A further consideration in dialysis patients. Semin Dial. 2019 Nov;32(6):485-489. doi: 10.1111/sdi.12834. Epub 2019 Jul 23. PMID: 31338891; PMCID: PMC6848753.
- Treatment of obesity and renal damage’ ‘Medications’: The authors referred to GLP-1 receptor agonists. Rather, sodium-glucose cotransporter (SGLT2) inhibitors are more relevant to obesity-related kidney diseases. They have to discuss SGLT2 inhibitors considering obesity.
We have expanded the information about SGLT2 (Page 14)
Pag 14. “Sodium-glucose cotransporter 2 inhibitors
Sodium-glucose cotransporter 2 inhibitor (SGLT2i) is a class of drug released in the last years with a mechanism that produce various beneficial effects in patients with diabetes, obesity and cardio and renal protection. Inhibition of glucose reabsorption in the proximal tubule produce glucosuria reducing the caloric burden, the sodium content with reduction of blood volume, and increasing the sodium arrival to the yuxtaglomerular corpus inhibiting the hyperactivity of the renal angiotensin system and reducing the filtration fraction given a protection to the kidney (99). As a consequence, slightly reduced weight, reduce BP and the GFR that result protector of renal function at large. Additional mechanisms in NH3, sympathetic activity and oxidative stress with beneficial effects complete a frame of a very useful drug. Several outcome trials supported the beneficial impact of the drug in the above mentioned conditions (100-103). The last European of Society of Cardiology and the European Society of Diabetes (ESC/EASD) recommended introduction of SGLT2i in the first step of diabetic patients with very high risk or previous Cardiovascular events (104). Then in patients with obesity associated a diabetes may be a good choice in order to protect renal function. The same in obese subjects with increased urinary albumin excretion or proteinuria. However, in patients with reduced GFR the efficacy of the drug is reduced and the impact in protection is diminished. In GFR <45ml/min/1.73m2 is almost negligible, it is a challenge their use. The beneficial impact observed in patients with GFR between 30-45ml/min/1.73m2 in trials and the absence of side effect points to their use off-label (105)”
- Janež A, Fioretto P. SGLT2 Inhibitors and the Clinical Implications of
Associated Weight Loss in Type 2 Diabetes: A Narrative Review. Diabetes Ther. 2021;12:2249-2261. doi: 10.1007/s13300-021-01104-z.
- Wanner C, Inzucchi SE, Lachin JM et al. Empagliflozin and progression of kidney disease in type 2 diabetes. N Engl J Med 2016;375:323–334
- Wiviott SD, Raz I, Bonaca MP et al. Dapagliflozin and cardio- vascular outcomes in type 2 diabetes. N Engl J Med 2019;380:347–357
- Neal B, Perkovic V, Mahaffey KW, de Zeeuw D, Fulcher G, Erondu N, Shaw W, Law G, Desai M, Matthews DR; CANVAS Program Collaborative Group. Canagliflozin and Cardiovascular and Renal Events in Type 2 Diabetes. N Engl J Med. 2017 Aug 17;377(7):644-657. doi: 10.1056/NEJMoa1611925. Epub 2017 Jun 12. PMID: 28605608.
- Perkovic V, Jardine MJ, Neal B et al. Canagliflozin and renal outcomes in type 2 diabetes and nephropathy. N Engl J Med 2019;380:2295
104 Cosentino F, Grant PJ, Aboyans V, Bailey CJ, Ceriello A, Delgado V, Federici M, Filippatos G, Grobbee DE, Hansen TB, Huikuri HV, Johansson I, Jüni P, Lettino M, Marx N, Mellbin LG, Östgren CJ, Rocca B, Roffi M, Sattar N, Seferović PM, Sousa-Uva M, Valensi P, Wheeler DC; ESC Scientific Document Group. 2019 ESC Guidelines on diabetes, pre-diabetes, and cardiovascular diseases developed in collaboration with the EASD. Eur Heart J. 2020;41(2):255-323. doi: 10.1093/eurheartj/ehz486.
- Li J, Fagbote CO, Zhuo M, Hawley CE, Paik JM. Sodium-glucose cotransporter 2 inhibitors for diabetic kidney disease: a primer for deprescribing. Clin Kidney J. 2019;12:620-628. doi: 10.1093/ckj/sfz100.
(8) Figures are smudgy. Correction of figures is necessary.
High resolution figures are available to the journal
Reviewer 2 Report
I found this review to be disjointed and I think it needs improvement in organization. I also think the review strayed away from its primary focus of obesity too often.
Specific comments
1) I think the beginning of the paper focus too much on epidemiology of CKD as a whole. The focus of the article should be obesity related kidney disease.
2) The authors quote a prevalence of CKD in the USA of 6.9%. What is the source of this prevalence? To my knowledge recent estimates of CKD in the USA are about 15% (using data found on CDC website).
3) The authors state that "BMI and CKD" are significant factors for development of CKD. I assume this is a typo.
4) The authors quote an odds ratio of 1.273 of "BMI for developing ESRD". It is not clear to me what this means. Is this a 27% increase by one point of a certain threshold? This needs to be explained further.
5) Table 1 discusses risk of CKD in individuals with metabolic syndrome. While I recognize that metabolic syndrome may be a consequence of obesity, this seems out of place. Metabolic syndrome has not been mentioned of defined yet in the manuscript. I think the epidemiology section should focus more on the relationship between CKD and obesity, rather than metabolic syndrome.
4) I would recommend the authors write terms rather than abbreviate them the first time they are used.
5) With respect to mechanisms, I would suggest the authors again focus on obesity itself rather than discussing other causes such as hyperglycemia and diabetes. I also think the authors should provide some of discussion of how some of the hormones they mention are affected by obesity.
6) Similarly with the effect of endothelial dysfunction/ nitric oxide- the authors note a number of conditions which cause endothelial dysfunction, but I think it would be interesting to know if any studies have linked obesity specifically with endothelial dysfunction. Similarly, does obesity itself influence NO? Are there any studies to that effect?
6) The comments about sarcopenia and measurement of body composition seem out of place in the clinical consequences section as they are not consequences of obesity.
7) I am confused on the authors discussion about albuminuria. Early in the manuscript they say it is rare, but later they seem to indicate that it can be a common finding in obesity-related glomerulopathy. I think this needs to be reconciled.
8) In the section on sub-nephrotic syndrome the authors state "this can be present in 30% of subjects". It is not clear to me what "this" refers to.
9) The sections on nephrolithiasis, renal replacement therapy, kidney transplant and renal cancer seem superficial, and the points discussed seem to have been arbitrarily decided. I would recommended omitting these sections unless the authors wish to expand them significantly. I also think these sections need to be better referenced if they do remain in the manuscript. It is also not clear what is meant by the statement that "for ever increment of 5 BMI units this risks increases 25%"
10) In the section on fatty kidney it is not clear what the statement "The role of blood pressure regulation and CKD has also be considered" means. Could the authors clarify this?
11) With respect to the treatment section, the authors state that losing weight is key by mitigating many of the bad outcomes that are associated with obesity. Are there any data that weight reduction actually does improve kidney outcomes? I think that would be important to include in this section.
12) With respect to lifestyle modifications, the authors say that a low calorie diet (with calorie restriction) is recommended. This recommendation should be referenced. Also is this a sustainable approach? Also, the authors quote some statistics associated with life-style measures, and these should be referenced.
13) While I think it is reasonable to use RAAS blockers in the setting of albuminuria, I would again consider the focus of the article is on obesity. Are there any data specifically looking at RAAS blockers in obesity per se? If not, I am not sure this belongs in this review.
14) Again with the drugs used to treat obesity, I would ask the question are there any data looking at weight loss and kidney outcomes, that is, are any data specific to obesity related kidney disease
15) In the conclusions the authors assert that BMI is the second most important factors in developing ESRD. What are the data for this assertion? This should be referenced.
Author Response
REVIEWER 2
Thanks to the reviewer for his detailed and useful review
I found this review to be disjointed and I think it needs improvement in organization. I also think the review strayed away from its primary focus of obesity too often.
Specific comments
(1)I think the beginning of the paper focus too much on epidemiology of CKD as a whole. The focus of the article should be obesity related kidney disease.
Specific articles to focus on obesity per se epidemiology are lacking as obesity is associated with comorbidities that are also causing CKD. We tried to answer why CKD despite improvement in treatment of T2D or hypertension is increasing, and obesity is one reason for that.
(2) The authors quote a prevalence of CKD in the USA of 6.9%. What is the source of this prevalence? To my knowledge recent estimates of CKD in the USA are about 15% (using data found on CDC website).
The source is the Ref 4.
(3) The authors state that "BMI and CKD" are significant factors for development of CKD. I assume this is a typo.
Yes, it was a typo. “BMI and diabetes”
4) The authors quote an odds ratio of 1.273 of "BMI for developing ESKD". It is not clear to me what this means. Is this a 27% increase by one point of a certain threshold? This needs to be explained further.
The sentence was fixed for a better compression
5) Table 1 discusses risk of CKD in individuals with metabolic syndrome. While I recognize that metabolic syndrome may be a consequence of obesity, this seems out of place. Metabolic syndrome has not been mentioned of defined yet in the manuscript. I think the epidemiology section should focus more on the relationship between CKD and obesity, rather than metabolic syndrome.
Obesity and metabolic syndrome are usually coexisted and obesity without metabolic abnormalities is discussed in the MHO paragraph but studies for obesity per se without concomitant diseases are limited
4) I would recommend the authors write terms rather than abbreviate them the first time they are used.
Terms have been described in each of the abbreviate them the first time used.
5) With respect to mechanisms, I would suggest the authors again focus on obesity itself rather than discussing other causes such as hyperglycemia and diabetes. I also think the authors should provide some of discussion of how some of the hormones they mention are affected by obesity.
How leptin and adiponectin are in obese subjects has been added in pag 7
6) Similarly with the effect of endothelial dysfunction/ nitric oxide- the authors note a number of conditions which cause endothelial dysfunction, but I think it would be interesting to know if any studies have linked obesity specifically with endothelial dysfunction. Similarly, does obesity itself influence NO? Are there any studies to that effect?
Studies have been added and commented in the text (references 51 to 53)
6) The comments about sarcopenia and measurement of body composition seem out of place in the clinical consequences section as they are not consequences of obesity.
The reviewer is true, but the comorbidity of sarcopenia jeopardized the assessment of obesity. The sentence have been modified.
Page 8. “At the time to estimate the association and the impact of obesity on renal disease, presence of sarcopenia, a no infrequent condition, can be misleading, since underestimate obesity (54)”.
7) I am confused on the authors discussion about albuminuria. Early in the manuscript they say it is rare, but later they seem to indicate that it can be a common finding in obesity-related glomerulopathy I think this needs to be reconciled.
We believe that across the manuscript references a albuminuria are frequent and always considered to be a frequent condition
8) In the section on sub-nephrotic syndrome the authors state "this can be present in 30% of subjects". It is not clear to me what "this" refers to.
The sentence has been fixed for better understanding.
Page 9. “This proteinuria at nephrotic levels can be present in 30% of subjects (63) in which there is sub-nephrotic proteinuria in the absence of edema, hypoalbuminemia and less hyperlipidemia”.
9) The sections on nephrolithiasis, renal replacement therapy, kidney transplant and renal cancer seem superficial, and the points discussed seem to have been arbitrarily decided. I would recommended omitting these sections unless the authors wish to expand them significantly. I also think these sections need to be better referenced if they do remain in the manuscript. It is also not clear what is meant by the statement that "for ever increment of 5 BMI units this risks increases 25%"
The objective of including the section of nephrolithiasis and renal cancer is just for alerting the reader about the possibilities that after bariatric surgery the risk of lithiasis increases y that a small but interesting association with cancer.
Concerning Renal replacement therapy and kidney transplant is to alert about the great impact that obesity in the relative small proportion of obese subjects that arrived to this stage of ESKD
Concerning the sentence it has been changed as well as the BB at consider that this is better
Pag 12: … and that for increment of 1 BMI units, the risk increases 6% (83)
10) In the section on fatty kidney it is not clear what the statement "The role of blood pressure regulation and CKD has also be considered" means. Could the authors clarify this?
The sentence has been removed since it is not clear and what tried to say is well present in the rest of the description.
11) With respect to the treatment section, the authors state that losing weight is key by mitigating many of the bad outcomes that are associated with obesity. Are there any data that weight reduction actually does improve kidney outcomes? I think that would be important to include in this section.
The manuscripts references (from 84 to 88) and the metanalysis (55) provide the information about the beneficial impact.
12) With respect to lifestyle modifications, the authors say that a low calorie diet (with calorie restriction) is recommended. This recommendation should be referenced. Also is this a sustainable approach? Also, the authors quote some statistics associated with life-style measures, and these should be referenced.
Pag 13. Reference 55
13) While I think it is reasonable to use RAAS blockers in the setting of albuminuria, I would again consider the focus of the article is on obesity. Are there any data specifically looking at RAAS blockers in obesity per se? If not, I am not sure this belongs in this review.
The impact of RAAS blockers in weight is minimal but the relevant protection of kidney and the reduction in insulin-resistance, frequent in obese subjects, merit to be mentioned.
14) Again with the drugs used to treat obesity, I would ask the question are there any data looking at weight loss and kidney outcomes, that is, are any data specific to obesity related kidney disease
Some trials with GLP-1 and less SGLT2i offered the beneficial impact in loosing weight, that contribute to the main outcome of the different studies referenced (100-103 in page 15)
15) In the conclusions the authors assert that BMI is the second most important factors in developing ESKD. What are the data for this assertion? This should be referenced.
The new references added in the section of Progression to CKD and ESKD, but in the sentence the word factor has been replace by marker.

Round 2
Reviewer 1 Report
The authors have adequately responded to my comments. However, there seems one misunderstanding in Table 1. I know that the previous studies provided the actual ORs (95% CIs) in each report. If I consult the references, I will know the exact values. Since this is a review article, the readers expect that they see and understand the results of previous reports at a glance. What I mean is to summarize the previous results using the actual figures as follows. The study endpoint: the onset of CKD or microalbuminuria. Median (or mean) follow-up period: (For example) 3.5 (interquartile range 2.5-5.1) years, if observational study. OR (95% CI): (For example) 2.60 (1.68- 4.03). I hope that this review article will be cited more than expected.
Author Response
The CI in the table has been added.
Reviewer 2 Report
1) Regarding the prevalence of CKD quoted in the manuscript- this was my error, I thought the authors were discussing CKD in general, I see now the authors were looking at CKD 3 and 4 only.
2) With respect to the albuminuria question, my concern is with the sentence "Clinically significant proteinuria is rare among the obese, but albuminuria may exist". I guess I am confused as to the difference between clinically significant proteinuria". Are the authors referring to nephrotic syndrome where there are symptoms? From a nephrologist's (at least this nephrologist's) perspective if individuals have proteinuria to a higher degree than microalbuminuria, that is clinically significant, although I agree it may not cause symptoms, but it can and does alter the clinical course and management decisions.
3) I'm still not clear on the section mentioned "odds ratio of quartile BMI". DOes this mean comparing the lowest quartile and the highest quartile. Similarly, in the section of renal cancer "and that for increment of 1 BMI
units, the risk increases 6%" my question is increment above what baseline level of BMI.
4) The sentence about 30% of subjects is still not clear to me.
"This proteinuria at nephrotic levels can be present in 30% of subjects [63] in which there is sub-nephrotic proteinuria in the absence of edema, hypoalbuminemia"
It am unclear if this is referring to nephrotic levels or sub-nephrotic levels.
5) My concern with the statement about BMI being the second most important marker wasn't so much about marker vs. factor. I still wonder why the authors assert the BMI is second most important. Where does this assertion come from?
Author Response
Reviewer 2
Thanks for your comments
1) Regarding the prevalence of CKD quoted in the manuscript- this was my error, I thought the authors were discussing CKD in general, I see now the authors were looking at CKD 3 and 4 only.
2) With respect to the albuminuria question, my concern is with the sentence "Clinically significant proteinuria is rare among the obese, but albuminuria may exist". I guess I am confused as to the difference between clinically significant proteinuria". Are the authors referring to nephrotic syndrome where there are symptoms? From a nephrologist's (at least this nephrologist's) perspective if individuals have proteinuria to a higher degree than microalbuminuria, that is clinically significant, although I agree it may not cause symptoms, but it can and does alter the clinical course and management decisions.
To avoid mistakes this has been modified and now is
Page 4: “Proteinuria of nephrotic range…”
3) I'm still not clear on the section mentioned "odds ratio of quartile BMI". DOes this mean comparing the lowest quartile and the highest quartile. Similarly, in the section of renal cancer "and that for increment of 1 BMI
units, the risk increases 6%" my question is increment above what baseline level of BMI.
The questions are clarified.
Page 2 ….with an odds ratio from the lowest to the higest BMI of 1.273 for developing ESKD, after adjustment for age, sex, systolic blood pressure, and proteinuria.
Page 8 … “The risk of kidney cancer was increased by 35% in overweight participants and by 76% in obese subjects in comparison to normal weight participants, irrespective of the gender”
4) The sentence about 30% of subjects is still not clear to me.
"This proteinuria at nephrotic levels can be present in 30% of subjects [63] in which there is sub-nephrotic proteinuria in the absence of edema, hypoalbuminemia"
It am unclear if this is referring to nephrotic levels or sub-nephrotic levels.
Page 6…Patients usually do not have proteinuria at the level of nephrotic syndrome in 30% of subjects [63] in which there is sub-nephrotic proteinuria in the absence of edema, hypoalbuminemia and less hyperlipidemia
5) My concern with the statement about BMI being the second most important marker wasn't so much about marker vs. factor. I still wonder why the authors assert the BMI is second most important. Where does this assertion come from?
We modify the sentence with a more explanation as follows:
Page 11… “BMI is the second most important marker in developing ESKD after proteinuria and one of the most relevant associated to the presence of CKD since obesity in frequently associated to hypertension, metabolic syndrome and diabetes”